# Charge Transfer Chromophores Derived from 3d-Row Transition Metal Complexes

**DOI:** 10.3390/molecules27238175

**Published:** 2022-11-24

**Authors:** Kira I. Pashanova, Irina V. Ershova, Olesya Yu. Trofimova, Roman V. Rumyantsev, Georgy K. Fukin, Artem S. Bogomyakov, Maxim V. Arsenyev, Alexandr V. Piskunov

**Affiliations:** 1Laboratory of Metal Complexes with Redox-Active Ligands, G.A. Razuvaev Institute of Organometallic Chemistry, Russian Academy of Sciences, 49 Tropinina Street, 603137 Nizhny Novgorod, Russia; 2International Tomography Center, Siberian Branch of the Russian Academy of Sciences, 3a Institutskaya Street, 630090 Novosibirsk, Russia

**Keywords:** catecholate, α-diimine, transition metal ion, chromophore, photoinduced intramolecular charge transfer, single-crystal X-ray diffraction, cyclic voltammetry, UV-vis-NIR spectroscopy, DFT study

## Abstract

A series of new charge transfer (CT) chromophores of “α-diimine-M^II^-catecholate” type (where M is 3d-row transition metals—Cu, Ni, Co) were derived from 4,4′-di-*tert*-butyl-2,2′-bipyridyl and 3,6-di-*tert*-butyl-*o*-benzoquinone (**3,6-DTBQ**) in accordance with three modified synthetic approaches, which provide high yields of products. A square-planar molecular structure is inherent for monomeric **[Cu^II^(3,6-Cat)(bipy*^t^*^Bu^)]∙THF** (**1**) and **Ni^II^(3,6-Cat)(bipy*^t^*^Bu^)** (**2**) chromophores, while dimeric complex **[Co^II^(3,6-Cat)(bipy*^t^*^Bu^)]_2_∙toluene** (**3**) units two substantially distorted heteroleptic D-M^II^-A (where D, M, A are donor, metal and acceptor, respectively) parts through a donation of oxygen atoms from catecholate dianions. Chromophores **1**–**3** undergo an effective photoinduced intramolecular charge transfer (λ = 500–715 nm, extinction coefficient up to 10^4^ M^−1^·cm^−1^) with a concomitant generation of a less polar excited species, the energy of which is a finely sensitive towards solvent polarity, ensuring a pronounced negative solvatochromic effect. Special attention was paid to energetic characteristics for CT and interacting HOMO/LUMO orbitals that were explored by a synergy of UV-vis-NIR spectroscopy, cyclic voltammetry, and DFT study. The current work sheds light on the dependence of CT peculiarities on the nature of metal centers from various groups of the periodic law. Moreover, the “α-diimine-M^II^-catecholate” CT chromophores on the base of “late” transition elements with differences in d-level’s electronic structure were compared for the first time.

## 1. Introduction

Originating in ancient time as a household craft [1], dyes chemistry has received well-deserved recognition and the widest industrial application, working on the diverse needs of society and making our life unthinkable without the existence of colorants (textile, food, cosmetics, paint and varnish manufacturing, etc.). At present, a search for new effective colorants (dyes or chromophoric systems), corresponding to newly emerging challenges, remains one of the most demanded chemical fields, which involves a numerous range of different technologies to generate and design a huge set of chromophoric compounds and materials [2].

In according with *K. Müllen and co-workers* [3], “functional dyes” should be considered rightfully as a pinnacle in a development of modern colorant chemistry: such dyes are intended to be implied in so-called “high-level photonic applications” for “smart” life, depending on the absorption/emission peculiarities of chromophores, e.g., photovoltaics (photosensitizers in solar cells) [4,5,6,7,8,9,10], optics (optical fibers, liquid-crystal materials) [11,12,13], luminescent-based (bioimaging, LED devices) [14,15,16,17,18], and electrochromic-based directions (e.g., “smart” mirror systems) [19,20,21,22,23,24,25,26,27,28] (references are mainly given on charge transfer complexes, see below).

The well-known and most effective to date chromophores can be classified [29] by their purpose, and by the mechanism of absorption/emission and ways to manage them [2]. In particular, avant-garde trends in working-out luminescent labels, which are indispensable in bioimaging, involve two types of fundamentally different chromophoric systems, quantum dots (QDs) and molecular dyes [29]. Unlike size-dependent absorption/emission of QDs, electron transition energy towards classical molecular colorants (e.g., organics, metal complexes) is available for a fine-tuning through the molecular design. Along with “resonant dyes” [29], displaying resonant absorption/emission owing to a delocalization of optical transitions over the molecular chromophoric system, the colorant with intramolecular charge transfer (CT dyes) deserve special attention from researchers. The sound advantages of CT chromophores [29] are the presence of high-intensive absorption/emission bands, as well as much larger Stokes shift against those for QDs.

The vast class of CT colorants is presented by transition metal complexes, the metal-to-ligand charge transfer (MLCT) of which is facilitated by non-planar molecular geometry [30,31] with high coordination numbers, and driven by a pronounced back donation phenomenon [32]. Remarkably, the molecular design of highly effective MLCT dyes derived from Mo [27], Os [4,6], Re [4,6,33], and mainly Ru [4,6,27,34,35] has been named an absolute priority, which can be explained by their use as photosensitizers (otherwise “redox mediators” [36]) in photovoltaic (PV) devices of Grätzel cell’s type [37,38]. As a rule, these MLCT chromophores demonstrate long-time stability under applied redox potential, as well as high photoelectric conversion efficiency (PCE) at ≈10% [6,39] in the case of third generation (3G) devices. Nevertheless, sound minuses of conventional Ru dyes are toxicity, high cost, and low abundance [40].

In this regard, CT chromophores of D-M^II^-A type (often this is dye with ligand-to-ligand charge transfer (LL’CT) and less coordination numbers) can provide a competitive alternative towards MLCT colorants due to noteworthy aspects: (1) pronounced charge separation on a molecular level with a planar mutual arrangement of interacting HOMO (donor organic part) and LUMO (acceptor organic part) [32]; (2) higher intense absorptivity; (3) an ability to fine-tune CT energy in the course of molecular design, varying electronic properties of ligands by functionalization; (4) lower energy of effective intramolecular CT in some cases, leading to an absorption/emission even in the NIR region [29] of an undeniable practical importance. It should be noted that a diminution in CT energy up to NIR absorption/emission is one of the priority tasks in the colorant chemistry. In particular, NIR light is ~45% of the solar energy [39] which makes the building of PV devices on the base of NIR photosensitizers reasonable (i.e., have the maximums of CT band in NIR region). In turn, the use of CT chromophores as luminescent labels in bioimaging screening requires NIR luminescence, since NIR radiation can penetrate into body tissues much more efficiently than UV/visible emission [18]. Additionally, an introduction of electrochromic NIR chromophores for electro-optic switching should be a promising project within fiber-optic technologies, especially those working at “telecommunication wavelengths” (from first to third wavelengths, 850, 1300, and 1550 nm, respectively) region [11].

Since a planar molecular geometry (most frequent in CT chromophores of D-M^II^-A type) is a favorable and necessary condition for realization of effective intramolecular LL’CT [32,41], at present the overwhelming majority of LL’CT chromophores comprises the derivatives of 10 group metals (Ni, Pd, Pt), which prefer a square-planar coordination environment [42]. Against 10 group metal CT dyes, very scant attention is paid to compounds of other transition elements, and a range of such chromophores is vastly limited in number and nature: thus, sporadic sources reported on 3d-row metal (Co [43] and Cu [44,45,46,47]) CT complexes. There are serious reasons for the observed disbalance. Many 3d-row elements tend to be surrounded by non-planar coordination environment, and the compounds often form with high coordination numbers (more than 4). In particular, tetrahedral polyhedron is an exemplary situation for four-coordinated Zn complexes [48,49,50,51], whereas in the case of iminoquinonato Cu derivatives, well-known equilibrium “square-planar—tetrahedron” [30,31] is finely susceptible towards ligands’ nature, i.e., bulkiness and carbon backbone rigidity, electronic properties and “hardness/softness” [52] of heteroatoms from potentially coordinating substituents. Regarding metals with more electron deficient d-orbitals, the tendency of complexes to increase their coordination number exists, and a steric factor plays a lesser role. For instance, quinonato/iminoquinonato Mn and Fe complexes are generally six-coordinated octahedral, while five-coordinated square-pyramidal/trigonal bipyramidal Mn and Fe derivatives are less common and contain more bulky iminoquinone type ligands [30,31], and the cases of four-coordinated Fe species are very scarce [53]. More complicated and an ambiguous situation is implemented for quinonato/iminoquinonato Co complexes: in contradistinction to exemplary six-coordinated quinonato derivatives, iminoquinonato species demonstrate a multeity among coordination numbers (4–6), that depends on slightest fluctuations in ligands’ bulkiness [30]. With a perspective to design CT chromophores of Co, great interest should be paid to coordinatively unsaturated Co complexes. An ability to generate stable square-planar derivatives was confirmed clearly by the example of iminoquinonato species [54,55,56,57,58], being driven by a steric factor. Moreover, the bright instance of tetrahedral Co complex and its isostructural Ni analog were reported recently [59]. Remarkably, a degree of a steric hindrance for coordinating atoms from ligands is an important factor within synthesis of D-M^II^-A chromophores, which determines their monomeric/oligomeric structure in the case of metals with the ability to form derivatives with higher coordination numbers than 4 (for instance, see ref. [47], as well as Section 2.2, and references in this chapter towards monomeric and oligomeric quinonato Cu/Co complexes).

As another serious omission in dye chemistry of CT chromophores of D-M^II^-A type, many available works take the main aim to perform a synthesis and routine analysis of general physic-chemical properties. On the other side, there is a scattered and non-systematized information concerning important CT characteristics, such as: (1) lifetime of excited state (all data for Pt^II^ chromophores) [10,16,17,60,61,62,63,64,65,66]; (2) CT energy in terms of its dependence on the nature of metal center [67,68,69,70,71]/electronic properties of ligands [11,17,28,32,41,42,60,61,62,63,64,68,70,71,72,73,74,75,76]/solvent polarity [32,42,60,61,62,63,64,67,76,77]. However, knowledge about these key parameters helps turn colorant into “functional dye”. For example, it should be highlighted that one comprehensive study of LL’CT energy [67] determined comparatively an influence of metal center’s nature: 3,5-di-*tert*-butyl-catecholato-M^II^-2,2′-bipyridine (where M is Ni, Pd, Pt) complexes display a slight increase of an extinction coefficient and narrowing of HOMO-LUMO gap within [Ni] < [Pd] < [Pt]. Remarkably, observed changes are expectedly minor and conditioned by an identical electronic structure of metal d-levels.

On the way to “functional CT dyes”, a number of recent reports can be identified as cutting-edge research designed to bridge the gap between “synthesis–structure–property” research pattern in attempts to apply or at least to reach/improve practically important characteristics. First of all, the extremely short lifetime of excited states of 3d-row transition metal complexes [40] can cross them off the set of solar cells’ photosensitizers and LED emitters. Being suitable model objects, Pt^II^ LL’CT dyes were chosen by many authors for lifetime’s measurements, including such methods, as time-resolved infrared (TRIR) and transient absorption (TA) spectroscopy [10,16,60,62,63,65,66]. Promising chapter within lifetime’s study, as well as within investigations of photoinduced electron spin polarization (ESP), belongs to magnetoactive Pd^II^ and Pt^II^ LL’CT chromophores, the catechol part of which is equipped by nitronylnitroxyl radical [78,79]. At the same time, an ability to generate long-lived excited species exists, for instance, in the case of dyes based on nodal metals with a full d-level. Such an approach was shown by example of *bis*(arylimino)acenaphthene Cu^I^ complexes [80]. The search for an alternative to the proven cisplatin can considered as a long-standing medical challenge, and is a typical for heteroligand Pt/Pd chromophores, mainly carboxylate [81,82,83,84,85] or quinonato [86,87,88,89,90] derivatives of Pt and Pd, which are augmented by ammine or condensed α-diimine moieties. On the other side, the heeds of modern fields such as solar energy have promoted the testing of high-effective LL’CT colorants in recent years, and the application of Ni [7,39,40,91], Pd [7,91], Pt [62,92,93,94], and Cu [36,80] complexes in dye-sensitized solar cells (DSSCs) is known.

Summarizing, the further molecular design of CT colorants should be continued as a prospective, targeted action, directed to a synergy between fundamental research and a response to practical challenges. Involving the derivatives of 3d-row “late” transition elements (first of all, Cu and Co) other than 10 group metals promises to be interesting: considerable results can be achieved, taking into account the diverse structure of d-levels, and lability in the formation of a coordination environment. Following above trend, we present CT chromophores **1**–**3** of 3d-row “late” transition metals (Cu, Ni, Co) based on 4,4′-di-*tert*-butyl-2,2′-bipyridyl and 3,6-di-*tert*-butyl-*o*-benzoquinone in terms of used synthetic approaches, molecular structures’ peculiarities, as well as CT characteristics, **Cu^II^(3,6-Cat)(bipy*^t^*^Bu^)**, **Ni^II^(3,6-Cat)(bipy*^t^*^Bu^)** and **[Co^II^(3,6-Cat)(bipy*^t^*^Bu^)]_2_** complexes, respectively. Interrelations between CT energy, structural features, and the nature of nodal metal center are explored in detail.

## 2. Results and Discussion

### 2.1. Syntheses of Complexes ***1***–***3***

Three different synthetic approaches (Figure 1) were chosen to obtain a series of new CT chromophores, monomeric **Cu^II^(3,6-Cat)(bipy*^t^*^Bu^)**, **Ni^II^(3,6-Cat)(bipy*^t^*^Bu^)** and dimeric **[Co^II^(3,6-Cat)(bipy*^t^*^Bu^)]_2_** complexes (**1**, **2** and **3** respectively).

An effective synthetic technique was proposed earlier [44,45,46,47] to obtain the “α-diimine-Cu^II^-catecholate” type compounds from a set of 1,2-diols (cathechol, 3,5-di-*tert*-butyl-catechol, tetrachlorocatechol, 3-*n*-nonylcatechol, 2,3-dihydroxynaphthalene), and α-diimines (ethylenediamine, 2,2′-bipyridyl, 1,10-phenanthroline, N,N,N′,N′-tetramethylethylenediamine), which differ in electronic properties (donor/acceptor ability) and in a bulkiness. A significant discrepancy of this method from one conventional for heteroleptic Ni^II^ analogs (see below) lies in an absence of a redox stage: derivative of the divalent metal ion (copper(II) sulfate) was chosen as a metal-containing compound, and dianion redox forms of donor ligands took part in the interaction. It should be noted that the similar approach is a common pattern for preparing of Pd^II^ and Pt^II^ LL’CT chromophores. In this case, additional reagents can be needed for an activation of the 1,2-diols. For instance, some authors [60,62,67,68,72,95] used bases, such as triethylamine, potassium *tert*-butoxide or sodium hydroxide. On the contrary, the synthesis of **Cu^II^(3,6-Cat)(bipy*^t^*^Bu^**) (**1**) was performed through a convenient “one-pot” redox process [96] with participation of neutral 3,6-di-*tert*-butyl-*o*-benzoquinone (**3,6-DTBQ**), 4,4′-di-*tert*-butyl-2,2′-bipyridyl, and an excess of metallic copper, yielding 81% of targeted product (Figure 1).

Due to the greatest prevalence of 10 group metals’ CT complexes (see Section 1), a set of well-developed synthetic techniques for such metal derivatives is known, depending on a redox state of *o*-quinone type ligand and an initial metal-containing compound. In particular, synthesis of the Ni chromophores proceeds in some cases with a participation of *o*-quinones, as well as Ni(0) derivatives coordinated by neutral ligands (tetracarbonyl nickel(0) [43] or *bis*(cyclooctadiene)nickel(0) [41,42], for example). The above method does not require any additional activating agents and involves a redox stage—a two-electron reduction of the redox-active *o*-quinone to a catechol form, that is accompanied by an oxidation of a metal center to a divalent state. Thus, following the authors [41,42], we have recently postulated [32] *bis*(cyclooctadiene)nickel(0) (**Ni(cod)_2_**) as a convenient “parent” to prepare the LL’CT chromophores of “α-diimine-Ni^II^-catecholate” type with high yields, applying neutral 3,6-di-*tert*-butyl-*o*-benzoquinone (**3,6-DTBQ**) and related 7,10-di-*tert*-butyl-2,5-dioxabicyclo(4.4.0)deca-1(10),6-diene-8,9-dione (**3,6-DTBQ^gly^**). In an absolute concordance with an analogous synthetic route described recently [32], the complex **Ni^II^(3,6-Cat)(bipy*^t^*^Bu^)** (**2**) was prepared, yielding 84% of a crystalline individual product. As can be seen from Figure 1, the first-step exchange interaction between **Ni(cod)_2_** and 4,4′-di-*tert*-butyl-2,2′-bipyridyl in ratio 1:1 proceeded in a glovebox under argon atmosphere, whereas the second redox stage leading to the final complex was carried out under anaerobic conditions.

Another activating method for the donor organic platform was applied by authors [43] to synthesize Co^II^ derivatives on the base of dry CoCl_2_, 3,6-di-*tert*-butyl-*o*-benzoquinone (**3,6-DTBQ**), 2,2′-bipyridyl or bulky substituted diazabutadienes: thallium catecholate was incorporated into the reaction mixture, being prepared in situ by a direct interaction between 3,6-di-*tert*-butyl-*o*-benzoquinone (**3,6-DTBQ**) and thallium amalgam (approximately 10-fold excess). Modifying slightly this method in the presented work, the dimeric **[Co^II^(3,6-Cat)(bipy*^t^*^Bu^)]_2_** complex was produced under anhydrous conditions with yield 78% in the course of two-step synthetic procedure, using sodium catecholate (**CatNa_2_**) (Figure 1).

As an interesting fact, the ability to manage the CT chromophoreʼs composition through a fine-tuning of the synthetic procedure (hydrous/anhydrous conditions), which allows generating monomer or dimeric species, was illustrated by the example of (3,5-di-*tert*-butyl-catecholato)(2,2′-bipyridine)copper(II) derivatives [47]. Moreover, an oligomerization can be considered as a frequent enough phenomenon for homoleptic transition metal derivatives of 3,5-di-*tert*-butyl-*o*-benzoquinone type platform: a dimerization of *bis*-*o*-benzosemiquinonato Cu^II^ moieties [97], as well as a tetramerization of *o*-benzosemiquinonato Co^II^ units [98] are fulfilled due to a dative bonding between oxygen atoms and metal ions. In contrast, a formation of monomeric molecular structures is a typical situation for coordination compounds derived from 3,6-di-*tert*-butyl-*o*-benzoquinone. A higher degree of the oxygen atom’s steric hindrance prevents the existence of oligomeric species. In this regard, an obtaining of dimeric **[Co^II^(3,6-Cat)(bipy*^t^*^Bu^)]_2_** complex (**3**), bearing two different bulky organic parts, is a noteworthy event. Summarizing, our strategy to prevent the oligomerization of D-M^II^-A parts by shielding of coordinating atoms from ligands due to an incorporation of substantially bulky *tert*-butyl groups turned out to be effective only in the Cu^II^ derivative **1**.

The synthesis of complexes **1**–**3** requires conditions excluding air moisture and oxygen. However, the resulting compounds are quite air stable and can be stored in open vessels in a crystalline form.

### 2.2. Molecular Structures of Complexes ***1***–***3***

It is acknowledged that the “α-diimine-M^II^-catecholate” type complexes have a square-planar structure. Hence, the planar mutual arrangement of overlapping HOMO and LUMO, facilitates an effective intramolecular ligand-to-ligand charge transfer [30,32,41]. As a consequence, to date, there is a numerous set of LL’CT heteroleptic chromophores, each of which is distinguished by a planar molecular structure [99].

In turn, complexes **1** and **2** were no exception. The coordination polyhedra possess a minimal distortion (Figure 2), in accordance with values τ_4_(**1**, **2**) = 0.05 [100,101]. Moreover, an ideal planarity (excluding *tert*-butyl groups) is featured for molecule of **1**, since the molecule lies on a special position. As a result, zero angles are formed by ligand planes, as well as by O(1)-Cu(1)-O(2) and N(1)-Cu(1)-N(2) planes of a coordination core. On the contrary, the peculiarities of molecular packing (detailed description is given below) cause a deviation from planarity in **2**: O(1A)-Ni(1A)-O(2A) and N(1A)-Ni(1A)-N(2A), O(1B)-Ni(1B)-O(2B) and N(1B)-Ni(1B)-N(2B) planes intersect at angles 2.81° and 3.53°, while the angles between catecholate dianion and bipyridine are found to be 11.96° and 13.98° (for molecules **A** and **B**, respectively).

Another molecular structure is inherent for the complex **3**: “α-diimine-Co^II^-catecholate” species are combined into dimer **[Co^II^(3,6-Cat)(bipy*^t^*^Bu^)]_2_∙toluene** (Figure 2) through a donation from the oxygen atoms of one D-M^II^-A part to the metal center of an adjacent moiety. As a result, each nodal Co^II^ ion is surrounded by an almost ideal (τ_5_ = 0.087 [102]) square-piramidal coordination environment with an apical oxygen atom from neighboring D-M^II^-A part of dimer. Corresponding distance Co(1)⋯O(2*) is equal to 2.0222(18) Å, which is less substantially for those towards cognate heteroleptic **[Cu^II^(3,5-Cat)(bipy)]_2_** [47] and homoleptic **[Cu^II^(3,5-SQ)_2_]_2_** [97] dimers, however comparable for bridging lengths in tetrameric **[Co^II^(3,5-SQ)_2_]_4_** [98] derivative (Table 1) of octahedrally coordinated Co^II^ ion (where **3,5-SQ** is 3,5-di-*tert*-butyl-*o*-semiquinone, **bipy** is 2,2′-bipyridine). In turn, intermetallic contacts Co(1)⋯Co(1*) = 2.8847(7) Å are shortened sizably, as follows from Table 1. In general, a formation of tetrameric structures is preferable for Co^II^ compounds derived from *o*-benzoquinone type ligands, as 3,5-di-*tert*-butyl-*o*-benzoquinone [98] and 9,10-phenanthrenequinone [103], but a turn to 3,6-di-*tert*-butyl-*o*-benzoquinone with a higher degree of oxygen atoms’ shielding leads usually to monomeric species [104]. However, the complex **3** is quite remarkable: the presence of two bulky *tert*-butyl groups in both coordinating ligands does not interfere a dimerization. Each chromophoric D-M^II^-A part in **3** is non-planar. Interligand angle values are sufficiently large (56.97°), almost twice those (32.44°) for **[Cu^II^(3,5-SQ)(bipy)]_2_** dimer mentioned above.

Metal complexes **1**–**3** under investigation possess the same unambiguous electronic structures of “α-diimine-M^II^-catecholate” species (Table 2). In particular, the classical bond lengths for dianionic O-donor ligand platforms (average C–C = 1.39–1.41 Å, C–O = 1.32–1.39 Å [31,105]) are found, with typical relative shortening for C(4)–C(5) distances of catecholate rings, due to a pronounced bonding nature for HOMO between mentioned carbon atoms [105]. In turn, a donor-acceptor coordination mode is implemented for neutral bipyridine organic moieties. Corresponding M-N and C-N bonds (Table 2) are in a good accordance with relative ones in known CT chromophores [32,68,72,87,89].

The “metal-heteroatom” lengths for Ni^II^ complex **2** are almost identical to those presented for **Ni^II^(3,6-Cat)(bipy)** analog [32]. In turn, the same distances in **Cu^II^(3,5-Cat)(bipy)** [47] slightly exceed ones for Cu^II^ derivative **1** (Table 3). A non-equidistance is observed for Co-O lengths owing to dimerization with a participation of oxygen atoms. As expected, the Co-O and Co-N bonds in **3** are elongated substantially compared to **Co^II^(3,6-Cat)(DAD)** (Table 3). This fact is in a good agreement with a growth in the ionic radius of cobalt as the coordination number increases [106], as well as with a higher steric hindrance of metal center in complex **3**.

The THF molecule in crystal **1** is disordered over two positions and is oriented by the oxygen atom to the hydrogen atoms of the bipyridine ligand. As a result, weak intermolecular hydrogen interaction O(THF)⋯H(16A) (2.653 Å) or O(THF)⋯H(19A) (2.600 Å) is realized.

An ideal lamellar crystal structure without π-stacking is inherent for complex **1** (Figure 3). The distances between molecular layers are equal to 4.559 Å. A similar situation with weakly interacting molecular layers was observed for a series of Pd^II^ compounds on the base of perfluoroalkyl-bearing catechol and various substituted bipyridines [68]. In contrast, the stabilization of a crystal structure is characterized for less bulky **Ni^II^(3,6-Cat)(bipy)** reported recently [32] through intermolecular π-π contacts of 2,2′-bipyridines. Indeed, an existence of π-stacking can be assumed as a distinctive feature for many lamellar CT chromophores, containing conjugated organic platforms with a rigid carbon backbone [32,68,72,107,108]. Nevertheless, the presence of bulky *tert*-butyl groups in donor/acceptor ligands, as well as solvate THF molecules (one molecule per molecule of complex) in **1** makes any π-π interactions impossible. Molecules of complex **1** in odd layers are strictly under each other, and a similar pattern is observed for molecules of even layers. So, the complex **1** demonstrates a mutual molecular arrangement, minimizing a repulsion of *tert*-butyl moieties (Appendix A).

Despite the lamellar laying realized in **1**, in general, the herringbone-like packing folded from T-motifs become the most preferable “building blocks” for a majority of CT chromophores of D-M^II^-A type with hindered organic moieties [32,67,73,87,89,90,95,109]. As a rule, molecules within such T-motif are stacked through the π-π interplays involving ligand orbitals, while there are no substantial π-π inter-stack contacts because of a steric shielding effect from bulky substituents in ligands (for instance, *tert*-butyls [95], *iso*-propylphenyls [73], perfluoroalkyls [68], etc). Complex **2** demonstrates the same pattern. The crystal structure contains isolated pairs of crystallographically independent molecules **A** and **B** (Figure 3). These T-motifs have a mutual arrangement close to perpendicularity, since molecular planes form angles at 72.31°/77.82°/83.37° (between molecules from different pairs—(**A** and **A**)/(**A** and **B**)/(**B** and **B**), respectively. Thus, the shortest contacts in dimeric pair are found: (1) between centroids of bipyridine ring *Cd*_bipy_ and chelate cycle *Cd*_Ch_, which is adjacent to catecholate dianion (*Cd*_Ch(**A**)_⋯*Cd*_bipy(**B**)_ 3.735 = Å, *Cd*_Ch(**B**)_⋯*Cd*_bipy(**A**)_ 3.783 = Å); (2) between nickel atoms (Ni(1A)⋯Ni(1B) 3.635 Å), Appendix A).

The molecular packing of compound **3**, continuing the tendencies for derivatives **1** and **2**, is organized according to the principle of least repulsion between complex’ molecules and solvate toluene molecules (Figure 3). The plane of the C_6_-ring of the catecholate and the NC_5_-ring of the bipyridine ligand of the neighboring molecule are almost perpendicular to each other. The corresponding dihedral angle is 74.68°. Thus, the possibility of realizing the π-π interaction is completely excluded. However, the shortest contact *Cd*_Cat_⋯C(23) = 3.695 Å (where *Cd*_Cat_ is centroid of C_6_-ring, Appendix A) may indicate the presence of intermolecular C-H⋯π interaction in the crystal of **3**.

It should be noted that an incorporation of sterically hindered functions into the ligands can be assumed as a rational strategy to increase the solubility of the complex. A tendency to π-π intermolecular stacking is weakening. For instance, a problem of solubility has been raised for Cu^II^ CT chromophores [44]. As a result, all studied coordination compounds **1**–**3** possess good solubility in a wide range of solvents with various polarity.

### 2.3. EPR Investigation for Complex ***1***

It is known that square-planar Ni^II^ complexes are characterized by a low-spin state of the metal ion. As a result, compound **2** is diamagnetic, as evidenced by a well-resolved NMR spectrum. A related Cu^II^ derivative **1** has the d^9^ configuration of a metal ion and is paramagnetic. X-Band EPR spectra recorded for solid sample of **1** (Figure 4a) demonstrate anisotropic signals with an axial symmetry (g_║_ = 2.18, g_┴_ = 2.08, A_║_(^63,65^Cu) = 220 G), which is characteristic for square-planar Cu^II^ compounds [110]. The finely resolved EPR spectrum for complex **1** in CH_2_Cl_2_ solution at room temperature also exhibits a four-line pattern (Figure 4b) as the hyperfine splitting (*hfs*) of unpaired electron with copper magnetic isotopes (I = 3/2). High-field components of spectrum show additional quintets 1:2:3:2:1 which are the result of *hfs* on two equivalent nitrogen nuclei ^14^N (I = 1) of bipyridine ligand. Spectrum parameters are g_i_ = 2.100, a_i_ (^63,65^Cu) = 92 G, a_i_ (2 ^14^N) = 10 G. No half-field signal is observed for either solid or vitrified (T = 150 K) CH_2_Cl_2_ matrix. This indicates the absence of any dimerization of **1** in crystals or in solution.

Dimeric compound **3** has two Co^II^ paramagnetic ions. However, the cobalt derivative does not have any X-band EPR signal in solid or in solution at room temperature and being frozen down to 100 K. This illustrates the preservation of the dimeric structure in the solution of **3**. The formation of a low-spin Co^II^ (S_Co_ = 1/2) complex would be observed with dissociation. Such Co^II^ derivatives show the corresponding signal in the EPR spectrum of frozen solutions [43].

### 2.4. Magnetochemical Study for Complex ***3***

An almost ideal square-pyramidal coordination environment of metal centers in **3** determined their high-spin state (S_Co_ = 3/2), being a usual situation for transition elements in d^7^ electronic configuration [30]. According to the temperature-variable magnetic susceptibility measurements, the high-temperature (T = 300 K) effective magnetic moment (μ_eff_) value for complex **3** is equal to 6.39 μ_B_, which transcends insignificantly the theoretical spin only value of 6.30 μ_B_ for two high-spin Co^II^ ions (S_Co_ = 3/2, quadruplet spin state) with g = 2.30. The magnitude of μ_eff_ changes slightly in the temperature range 300–100 K, whereas below 100 K the μ_eff_(T) curve drops sharply down to 0.83 μ_B_ at 2 K, that clearly indicates a presence of antiferromagnetic exchange interactions between spins of two Co^II^ ions (Figure 5). Analysis of the μ_eff_(T) dependence using dimer model (Spin Hamiltonian H = –2J·S_1_S_2_) allows estimating exchange interaction parameter J. The best fit values of g-factor and J are 2.39 ± 0.01 and −4.76 ± 0.03 cm^−1^, respectively.

### 2.5. Electrochemical Study for Complexes ***1***–***3***

The electrochemical behavior of Cu^II^ and Ni^II^ monomeric derivatives **1** and **2** can be considered as similar, consisting in ligand-centered redox-stages (Figure 6): reversible one-electron oxidative wave corresponds to a transition “catecholate dianion → semiquinone radical anion”, and quasi-reversible one-electron reduction of coordinated bipyridine ligand is observed.

It should be noted that the values of electrochemical potentials for complex **2** and Ni^II^ analog **Ni^II^(3,6-Cat)(bipy)** with unsubstituted 2,2′-bipyridine [32] are very close. However, the HOMO → LUMO gap is wider expectedly for **2** due to the presence of electron-donor *tert*-butyl fragments in acceptor diimine part (Table 4).

On the contrary, dimeric Co^II^ complex **3** displays another character of voltammogram (Figure 6). Quasi-reversible one-electron redox process, which is detected with significant displacement towards reductive steps for **1** and **2**, can be attributed to Co^II^ → Co^I^ reduction. Noticeably, the same metal-centered redox transformations were described in detail for Co^II^ derivatives based on the polypyridyl multidentate ligand platforms [111,112]. Two consecutive one-electron reversible peaks are observed on the oxidative curve of the voltammogram. These processes can be attributed with equal probability to both the oxidation of catecholate ligands to *o*-semiquinone ones, and to the transition of Co^II^ → Co^III^ [112].

### 2.6. UV-vis-NIR Spectroscopy for Complexes ***1***–***3***

As it was highlighted above, a comparison of chromophores from a standpoint of CT energy is a quite rare final goal for corresponding investigations. *H.-C. Chang and co-workers* have shown [67] that changes in absorptivity among analogous CT dyes on the base of 10 group metals (Ni, Pd, Pt) cannot be considered as “substantial”: the wavelength shift of LL’CT band did not transcend at ~20 nm (in the same solvent) between derivatives of metals from adjacent periods. On the other side, to the best of our knowledge, there are no targeted investigations establishing how the nature (d-level structure, hence a coordination sphere volume and preferable coordination environment) of metals from other groups affects CT energy. However, this task may represent certain perspectives within CT chromophores’ development and design. Thus, in this paper, we will try to work on this issue.

The key feature of absorption spectra for complexes **1**–**3** is the presence of high-intense broadened CT bands extending in the visible/NIR regions (Figure 7), and a pronounced negative solvatochromic effect (Table 5).

Regarding the Ni^II^ complex **2**, the energetic parameters of LL’CT (according to CV data, see above) are identical to those for the related Ni^II^ chromophore, containing 3,5-di-*tert*-butyl-catecholate ligand instead of 3,6-substituted counterpart (Table 5). An incorporation of *tert*-butyl functions to 2,2′-bipyridine platforms in **2** expectedly initiates a slight hypsochromic shift (within 20–35 nm in dependence on solvent) of the LL’CT bands (Table 5) in comparison with recently published (3,6-di-*tert*-butyl-catecholato)(2,2′-bipyridine)nickel(II) [32]). As can be seen from the current results (see Table 5), an equivalent effect towards LL’CT (according to CV data, see above) energy can be caused by a replacement of the metal center from Ni^II^ to Cu^II^ ion with the unchanged ligands and a similar molecular structure (monomer, square-planar coordination core): the LL’CT bands’ shifts to shortwave region are equal to 26–40 nm in the case of Cu^II^ analog **1** (Table 5). Another important fact that can be seen when comparing the photophysical characteristics of isostructural complexes **1** and **2** is that replacing the Ni^II^ cation with a Cu^II^ almost halves the extinction coefficient.

It is necessary to note that compounds **1**, **2,** and abovementioned metal complexes reported earlier [32,67] can be classified as so-called “visible CT dyes (i.e., have the maximums of CT band in visible region)” based on λ_max_ values (without considering the band width). The generation of effective NIR chromophores is extremely relevant and dictated by different challenges, thereby one of the prior strategies within molecular design of CT dyes should be considered a targeted decrease of CT energy until to NIR absorptivity. One way is a modification of the donor catecholate moiety against acceptor diimine part. This is a quite efficient approach: (1) a slight turn in electronic properties of 3,6-di-*tert*-butyl-catecholate ligand in the course of an annelation with electron-donor glycol fragment shifted the LL’CT band at 95 nm, producing NIR dye (λ_max_ = 810 nm, in toluene solution) of Ni^II^ (Table 5) [32]; (2) a cardinal design (*A. F. Heyduk with co-workers*) [42]) of catecholates’ composition and structure (3,5-di-*tert*-butyl-*o*-benzoquinone, tetrachloro-1,2-quinone and 9,10-phenanthrenequinone were used) among a wide set of “α-diimine-Ni^II^-catecholate” species resulted a record-breaking red LL’CT band’s displacement at about ~430–550 nm up to deep NIR region (the largest value λ_max_ = 1370 nm, in THF solution).

Here, we report another possible strategy of molecular design on the way to NIR CT chromophores: a turn from 10 group metal in Ni^II^ complex **2** (convenient standard in LL’CT chemistry due to a preference of a square-planar coordination polyhedron) to the 3d-row transition element of different nature with more labile coordination sphere. Dimeric Co^II^ derivative **3** demonstrate UV-vis-NIR spectra with high-intense (within studied complexes **1**–**3**) (M + L)L’CT (according to CV data, see above) absorption bands which are shifted from the visible to the NIR region (Table 5). It should be remarked especially that the dimeric structure of chromophore **3** is maintained regardless of the polarity and coordination ability of used solvents (CH_3_CN, DMF, CH_2_Cl_2_, THF) that is proved by a synergy of UV-vis-NIR and EPR-spectroscopy data: (1) the general character of UV-vis-NIR spectra was independent towards concentration effect, as well as solvent nature—high-intense (M + L)L’CT bands are presented (Figure 7); (2) no X-band EPR signal for complex **3** was observed in CH_2_Cl_2_ solution. Obviously, dimer destruction should generate “α-diimine-Co^II^-catecholate” species, and an implementation of a square-planar or tetrahedral molecular geometry is equiprobable for similar four-coordinated cobalt derivatives, as highlighted in Section 1. The tetrahedral mutual arrangement between HOMO and LUMO eliminates CT and, therefore, an observation of corresponding CT band in UV-vis-NIR spectra is impossible. In turn, square-planar coordination environment provides a low spin state of divalent cobalt ion (S_Co_ = 1/2) [30,31], conditioning the paramagnetism of such “α-diimine-Co^II^-catecholate” monomer. Thus, the existence of a dimeric structure for **3** in solution is beyond doubt.

The pronounced negative solvatochromism is established for chromophores **1**–**3** as expressed by a fine linear dependence *E*_T_^N^(λ) (Figure 8) between CT bands and the normalized empirical parameter *E*_T_^N^ by Dimroth and Reichardt [113]. The *E*_T_^N^ parameter is obtained empirically on the base of the solvent-dependent CT band’s shift for the standard N-phenoxypyridinium betaine dye, and, hence, describes the non-specific solvent polarity. CT bands’ displacement for chromophores **1**–**3** is explained by fine-sensitive energy of non-polar excited states towards the solvent polarity. So, a turn from CH_3_CN to THF or toluene facilitates CT, causing substantial red shifts (Table 5) due to CT energy’s decrease. The found parameters of solvatochromic displacement are comparable with those for known “α-diimine-Ni^II^-catecholate” analogs (Table 5), being one of the largest reported values [32,42,67].

### 2.7. DFT Calculations

Quantum chemical calculations by the Density functional theory (DFT) method at the B3LYP/6-311++g(2d,2p) level of theory were performed to study the electronic structure of complexes under investigation. The experimental (X-ray) geometries were used as the starting points and optimized. Compound **1** was calculated in an open-shell approximation with S = 1/2 ground state. EPR spectroscopy data confirms doublet spin state for Cu^II^ derivative **1**. Complexes **2** and **3** were calculated in a closed-shell approximation with S = 0 ground state. The diamagnetic nature of Ni^II^ complex **2** was evidenced by well-resolved NMR spectrum with no line broadening. The singlet ground state for dimeric Co^II^ derivative **3** is clear from the solid state magnetochemical measurements. The selected frontier orbitals are presented in Figure 9 and Figure 10.

The LUMO orbital (−2.51 eV) (Figure 9, (**4**)) is located mainly on the acceptor diimine part of the Ni^II^ complex **2**. This well-defined HOMO (−4.38 eV) (Figure 9, (**2**)) occupies catecholate ligand with small nickel d_xy_ and bipyridine system contribution. Orbitals energy and the nature of their distribution is quite obvious for such type of complexes [32]. The energy of the HOMO orbital increases significantly (0.41 eV) when *tert*-butyl substituents are introduced into the 2,2′-bipyridine [32]. These changes lead to an increase in the HOMO-LUMO energy gap (1.87 eV) and a corresponding hypsochromic shift observed in the electronic spectrum of compound **2** compared to the previously published derivative containing unsubstituted bipyridine. The energy (1.82 eV) of longwave absorption band (680 nm in toluene) is in a good agreement with calculated HOMO-LUMO energy gap. The nearest orbitals with a prevailing contribution of metal AO are HOMO-2 (d_z2_, −6.19 eV) and LUMO+3 (d_x2-y2_, −1.14 eV). Thus, complex **2** can be considered as a LL’CT donor-acceptor chromophore which is controlled by the ligand framework.

The situation with the formation of frontier orbitals for the related Cu^II^ complex **1** turns out to be very similar. HOMO (−4.28 eV) and α-LUMO (−2.58 eV) are located on the donor catecholate and acceptor diimine ligands respectively (Figure 9, (**1**) and (**3**)). The appearance of an additional electron at the d-level of the metal leads to a general decrease in the energy of metal atomic orbitals. Thus, the highest for squire-planar configuration semioccupied d_x2-y2_ orbital is α-HOMO-2 (−5.93 eV) for complex **1**. This orbital is responsible for the doublet spin state of the compound (Figure 9, (**5**)). A decrease in the energy of the d-orbitals in the complex **1** (compared to the Ni^II^ derivative **2**) leads to a decrease in the contribution of the d_xy_ orbitals in the HOMO and LUMO. It reduces the efficiency of charge transfer between the donor and acceptor fragments of the molecule and the molar extinction coefficient for the Cu^II^ compound **1** is about two times lower than the Ni^II^ analog **2**.

The composition and energy of the frontier orbitals undergo a significant change in the Co^II^ complex **3** (Figure 10). HOMO (−3.82 eV) and LUMO (−2.25 eV) have a prevailing metal character which is confirmed by the data of electrochemical studies. A noticeable contribution of the diimine fragment is observed for LUMO. The nearby HOMO-1 (−4.19 eV) and HOMO-2 (−4.44 eV), as well as LUMO+1 (−2.00 eV) and LUMO+2 (1.97 eV), are occupied by catecholate and bipyridine ligands respectively. These orbitals have an appreciable contribution of cobalt d-orbitals. Thus, the cobalt ion in the resulting donor-acceptor complex provides effective charge transfer due to the active involvement of metallic d-orbitals into the frontier orbitals formed by redox-active organic fragments. It should be noted that the decrease in the molar extinction coefficient in low-polar solvents for Co^II^ derivative **3** is not observed, despite the formation of a five-coordinate environment of the metal atom. This configuration of the complex can provide charge transfer between the donor and the acceptor [114]. However, it is significantly inferior to structures with a square-planar geometry.

## 3. Materials and Methods

The comprehensive general information towards used materials and methods is collected in Appendix A.

### Syntheses and Characterization of Complexes ***1***–***3***

To prepare complexes **1**–**3**, combined conditions were implemented: the second stages of the two-step synthetic procedure in the case of compounds **2** and **3**, as well as the synthesis of **1**, were performed under Schlenk line, while an argon atmosphere glovebox was used to carry out the second stages for **2** and **3**.

**Complex 1·THF**, **[Cu^II^(3,6-Cat)(bipy^tBu^)]∙THF**. Equimolar amounts of 3,6-di-*tert*-butyl-*o*-benzoquinone (**3,6-DTBQ**) (0.5 g, 2.27 mmol) and 4,4′-di-*tert*-butyl-2,2′-bipyridyl (0.61 g, 2.27 mmol) reacted with an excess of sliced copper foil in dry THF (20 mL) in evacuated ampoule. The resulting solution, which turned from bright-green to ink-blue color, was decanted from copper and partially evaporated, after that dark fine-crystalline product formed. Precipitate was collected by filtration and washed with THF under reduced pressure. Single crystals of **[Cu^II^(3,6-Cat)(bipy*^t^*^Bu^)]∙THF** with C_36_H_52_CuN_2_O_3_ composition, which are suitable for X-ray diffraction experiment, were obtained from THF through a slow evaporation of solution under residual pressure. The total yield of product with C_36_H_52_CuN_2_O_3_ formula is 1.15 g (81%) based on the starting ligands.

The polycrystalline samples of **1** for an elemental analysis, IR-, EPR-, UV-vis-NIR spectroscopy and cyclic voltammetry were prepared from THF.

IR (Nujol, KBr) cm^−1^: 488 (m), 538 (w), 557 (m), 604 (s), 650 (s), 698 (s), 742 (m), 752 (w), 771 (m), 809 (m), 851 (s), 899 (s), 923 (m), 939 (s), 980 (s), 1023 (m), 1034 (m), 1053 (s), 1117 (m), 1145 (m), 1172 (w), 1202 (m), 1246 (s), 1258 (s), 1279 (m), 1304 (w), 1347 (m), 1367 (m), 1409 (s), 1552 (s), 1582 (w), 1620 (s), 1706 (w), 1787 (w), 1832 (w), 1965 (w), 3062 (m), 3114 (w).

Anal. Calc. for the complex **[Cu^II^(3,6-Cat)(bipy*^t^*^Bu^)]∙THF** with C_36_H_52_CuN_2_O_3_ composition (%): C, 69.25; H, 8.39; N, 4.49. Found: C, 69.56; H, 8.86; N, 4.75.

**Complex 2**, **Ni^II^(3,6-Cat)(bipy^tBu^)**. A colorless solution of 4,4′-di-*tert*-butyl-2,2′-bipyridyl (0.5 g, 1.86 mmol) in dry toluene (20 mL) was poured to an equimolar yellow solution of Ni(cod)_2_ (0.51 g, 1.86 mmol) in the same solvent (20 mL) in glovebox. A resulting reaction mixture was stirred in an evacuated ampoule for an hour under an ambient temperature when a deep purple solution was obtained. An addition of an equimolar green solution of **3,6-DTBQ** (0.41 g, 1.86 mmol) in dry toluene (10 mL) changed a reaction mixture’s color to ink blue. A slow removal of the reaction mixture under reduced pressure facilitated a growth of ink rectangular-shaped single crystals of **Ni^II^(3,6-Cat)(bipy*^t^*^Bu^)** with C_32_H_44_N_2_NiO_2_ composition. The polycrystalline precipitate was filtered off and washed with dry hexane under residual pressure. The total yield for product **Ni^II^(3,6-Cat)(bipy*^t^*^Bu^)** with C_32_H_44_N_2_NiO_2_ formula is 0.84 g (84%) based on the starting ligands.

The polycrystalline samples of **Ni^II^(3,6-Cat)(bipy*^t^*^Bu^)** for an elemental analysis, IR-, ^1^H NMR, UV-vis-NIR spectroscopy, and cyclic voltammetry were obtained from toluene, and dried additionally in anaerobic conditions.

Anal. Calc. for the solvent-free complex **Ni^II^(3,6-Cat)(bipy*^t^*^Bu^)** with C_32_H_44_N_2_NiO_2_ composition (%): C, 70.21; H, 8.10; N, 5.12. Found: C, 69.83; H, 8.09; N, 5.00.

IR (Nujol, KBr) cm^−1^: 462 (m), 485 (w), 520 (m), 540 (m), 569 (w), 599 (s), 612 (m), 648 (s), 670 (w), 710 (s), 727 (s), 743 (m), 776 (m), 810 (m), 846 (s), 887 (m), 901 (m), 927 (w), 941 (m), 982 (s), 1024 (m), 1035 (m), 1076 (w), 1115 (w), 1141 (m), 1173 (w), 1201 (m), 1244 (s), 1263 (m), 1281 (m), 1295 (w), 1310 (m), 1348 (m), 1366 (m), 1412 (s), 1478 (m), 1546 (m), 1590 (m), 1620 (s), 1793 (w), 1804 (w), 1825 (w), 1954 (w), 3073 (m).

^1^H NMR (300 MHz, CDCl_3_, 20 °C, δ/ppm): 1.42 (*^t^*Bu, 18H), 1.47 (*^t^*Bu, 18H), 6.28 (CH_Ar_, 2H), 7.47 (CH_Ar_, 1H, *J* = 1.74), 7.49 (CH_Ar_, 1H, *J* = 1.88), 7.67 (CH_Ar_, 2H, *J* = 1.46), 8.73 (CH_Ar_, 2H, *J* = 5.99).

**Complex 3∙toluene**, **[Co^II^(3,6-Cat)(bipy^tBu^)]_2_∙toluene**. A colorless solution of 4,4′-di-*tert*-butyl-2,2′-bipyridyl (0.5 mg, 1.86 mmol) in dry THF (5 mL) was mixed with a blue suspension of an anhydrous CoCl_2_ (0.24 mg, 1.86 mmol) in the same solvent (10 mL) in glovebox, producing an ink-colored solution. Interaction between the resulting mixture and a pale solution (dry THF, 10 mL) of prepared in situ **CatNa_2_** (1.86 mmol) in evacuated ampoule facilitated an immediate turn to a green color of reaction. Solvent was changed to dry CH_2_Cl_2_ with a subsequent removing of formed white NaCl precipitate, after that the filtrate was stored in refrigerator for an hour. Obtained purple fine-crystalline product was separated by filtration in vacuum, and dried. The total yield for product **[Co^II^(3,6-Cat)(bipy*^t^*^Bu^)]_2_∙CH_2_Cl_2_** with C_65_H_90_Cl_2_Co_2_N_4_O_4_ formula is 1.41 g (78%) based on the starting ligands.

A hot toluene solution of complex was evaporated slowly under anaerobic conditions to grow single crystals (**[Co^II^(3,6-Cat)(bipy*^t^*^Bu^)]_2_∙toluene**, C_71_H_96_Co_2_N_4_O_4_ composition) suitable for X-ray diffraction measurements.

The synthesis in situ of **CatNa_2_** was carried out through the direct interaction of **3,6-DTBQ** with an excess of metallic Na in THF. Reagents were stirred for an hour in evacuated ampoule, while the bright-green initial color of the reaction mixture turned to pale. The corresponding technique is well-known and described earlier [115].

The polycrystalline samples of **3** for an elemental analysis, IR-, EPR-, UV-vis-NIR spectroscopy, cyclic voltammetry, and magnetic susceptibility measurements were prepared from CH_2_Cl_2_.

Anal. Calc. for the complex **[Co^II^(3,6-Cat)(bipy*^t^*^Bu^)]_2_∙CH_2_Cl_2_** with C_65_H_90_Cl_2_Co_2_N_4_O_4_ composition (%): C, 66.15; H, 7.69; N, 4.75. Found: C, 65.74; H, 7.87; N, 4.95.

IR (Nujol, KBr) cm^−1^: 486 (m), 501 (m), 539 (m), 546 (w), 559 (s), 608 (s), 652 (s), 676 (s), 696 (m), 742 (s), 760 (w), 785 (s), 804 (m), 847 (s), 862 (m), 887 (m), 900 (m), 922 (m), 931 (s), 972 (s), 1016 (m), 1080 (w), 1121 (m), 1144 (s), 1170 (w), 1192 (m), 1198 (w), 1214 (s), 1249 (s), 1265 (w), 1277 (m), 1346 (w), 1367 (m), 1394 (s), 1407 (m), 1478 (s), 1530 (m), 1551 (s), 1567 (w), 1614 (s), 1694 (w), 1748 (w), 1770 (w), 1823 (m), 1843 (w), 1948 (w), 1969 (w), 3046 (w), 3065 (m).

## 4. Conclusions

A novel strategy within a molecular design of CT chromophores aimed at the CT energy management was implemented by the example of new “α-diimine-M^II^-catecholate” CT dyes **1**–**3** (where M is 3d-row transition metals—Cu, Ni, Co): the varying of the metal center’s nature (electronic structure of d-level) was performed. As regards the preparation of CT dyes, three synthetic routes were applied and optimized, providing a high yield of complexes with different composition and molecular structure. So, the chosen approaches allow obtaining the square-planar monomeric LL’CT species (Cu^II^ and Ni^II^ derivatives **1** and **2**, respectively), as well as dimeric Co^II^ (M + L)L’CT chromophore **3** containing two “α-diimine-M^II^-catecholate” moieties united through the donor ability of oxygen atoms from catecholate dianions.

Corresponding changes in CT energy were investigated in detail by the synergy of experimental and theoretic tools, namely UV-vis-NIR spectroscopy, cyclic voltammetry, and DFT study. In particular, a considerable red CT band’s displacement at approximately 130–150 nm was achieved by the turn from Cu^II^ to Co^II^ center with the same ligand platforms (neutral 4,4′-di-*tert*-butyl-2,2′-bipyridyl and 3,6-di-*tert*-butyl-catecholate dianion). Moreover, a fine-tunable interrelation between CT energy and solvent polarity was established, expressed in a pronounced negative solvatochromic effect with substantial blue CT band’s shifts. Total values are equal to 153, 137, and 58 nm for complexes Cu^II^ (**1**), Ni^II^ (**2**), and Co^II^ (**3**), respectively (from toluene to CH_3_CN for **1**, **2**; from THF to CH_3_CN for **3**).

## Figures and Tables

**Figure 1 molecules-27-08175-f001:**
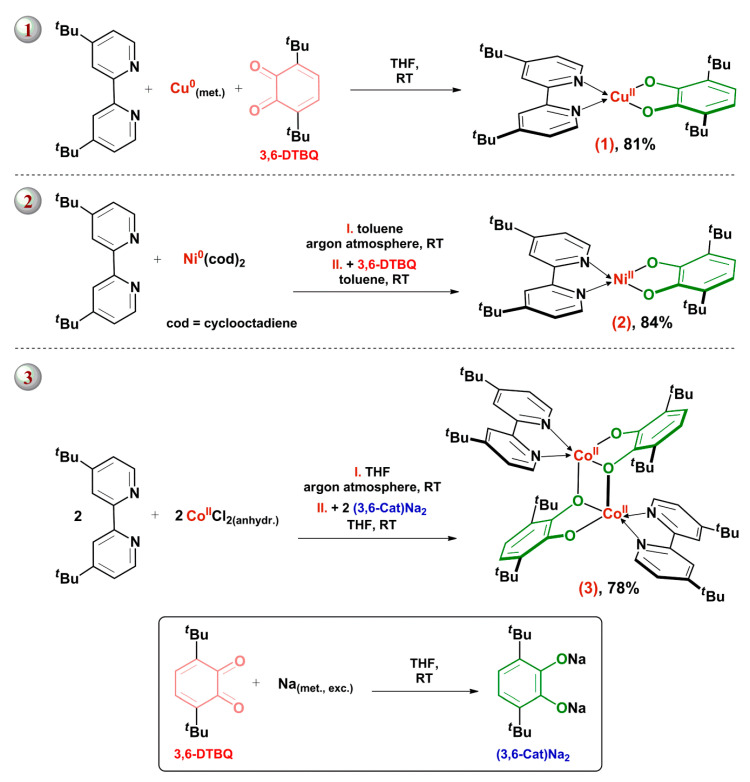
Synthetic routes for complexes **1**–**3**.

**Figure 2 molecules-27-08175-f002:**
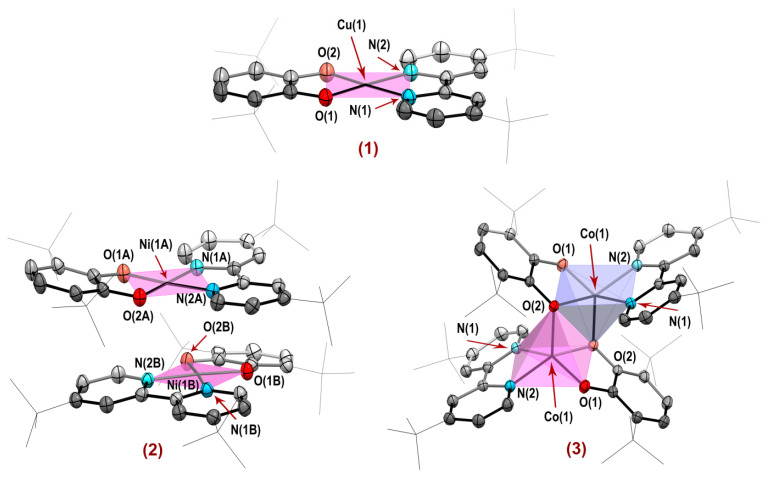
Views of (**1**), (**2**) and (**3**) molecular structures with 30% thermal probability ellipsoids. Hydrogen atoms are omitted for clarity.

**Figure 3 molecules-27-08175-f003:**
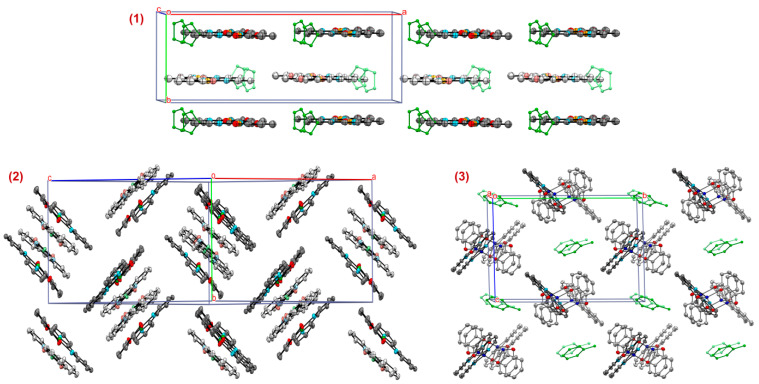
The fragments of crystal packings for complexes **1**–**3**. Faded views are given for: the molecules from even layer (in the case of compound **1**); the molecules **B** (in the case of complex **2**); one monomeric D-M^II^-A part (in the case of complex **3**). Structures are given with 30% thermal probability ellipsoids. Hydrogen atoms and *tert*-butyl groups are omitted for clarity.

**Figure 4 molecules-27-08175-f004:**
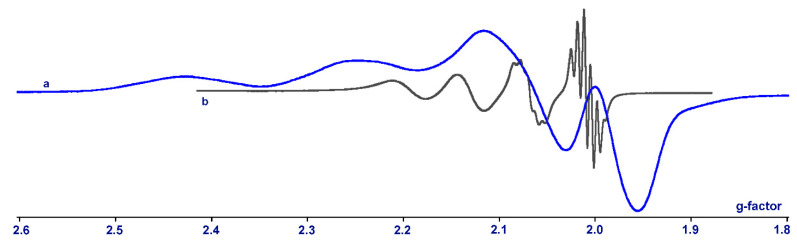
EPR spectra for complex **1** in solid state (**a**) and in CH_2_Cl_2_ solution (**b**) at room temperature.

**Figure 5 molecules-27-08175-f005:**
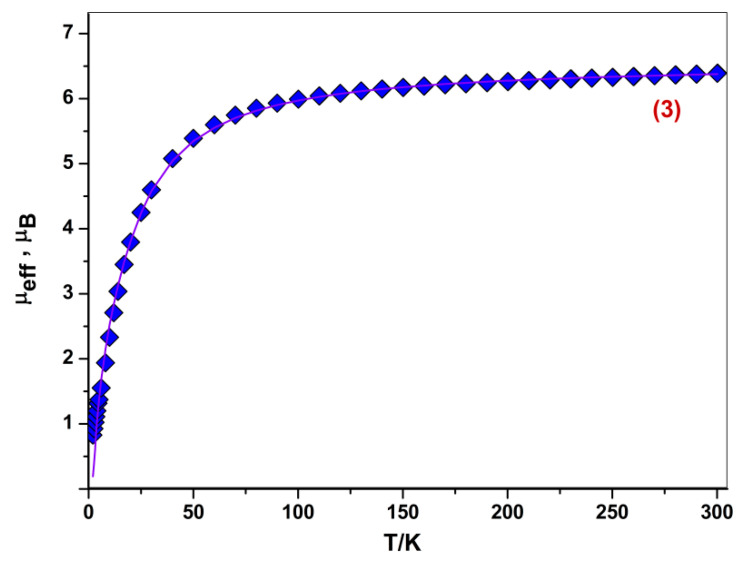
The temperature dependence of the effective magnetic moment μ_eff_(T) for complex **3**. Solid line is a theoretical curve.

**Figure 6 molecules-27-08175-f006:**
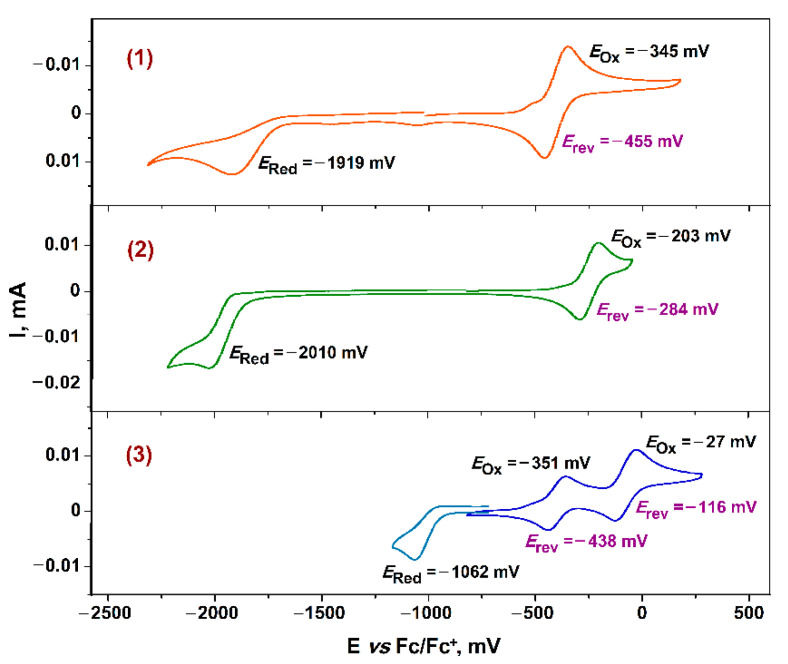
The CV data for complexes **1**–**3** (GC electrode, Ag/AgCl/KCl, 0.2 M Bu_4_NClO_4_, CH_2_Cl_2_, C = 2·10^−3^ mol/dm^3^, V = 0.2 V·s^−1^, Ar). Corrections to Fc/Fc^+^ couple as a standard were made.

**Figure 7 molecules-27-08175-f007:**
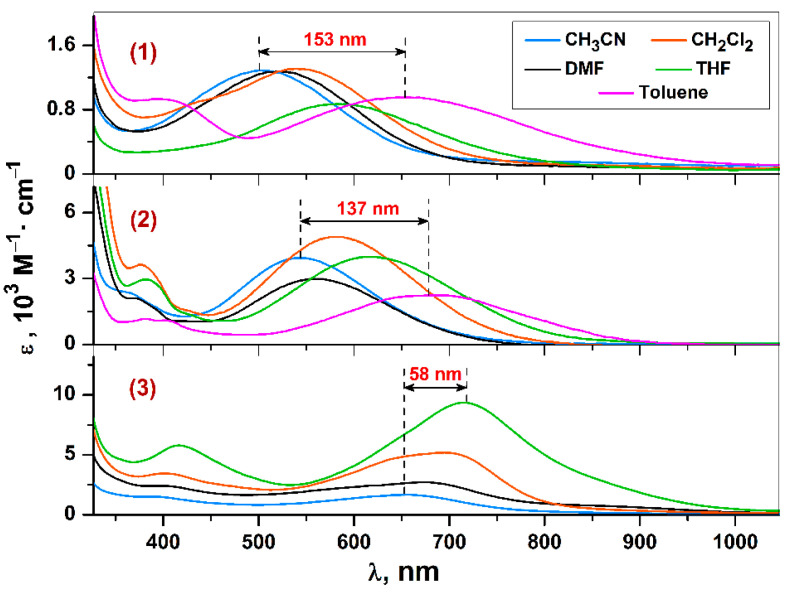
UV-vis-NIR spectra for complexes **1**–**3**.

**Figure 8 molecules-27-08175-f008:**
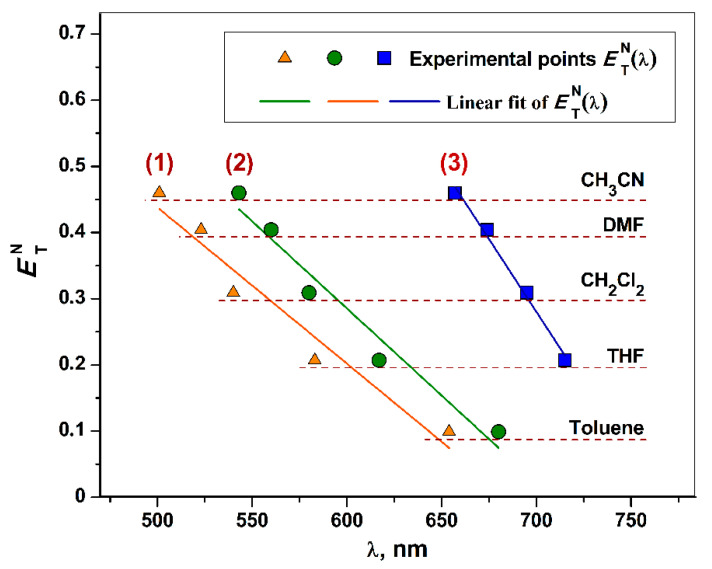
*E*_T_^N^(λ) dependences and linear fits for complexes **1**–**3**.

**Figure 9 molecules-27-08175-f009:**
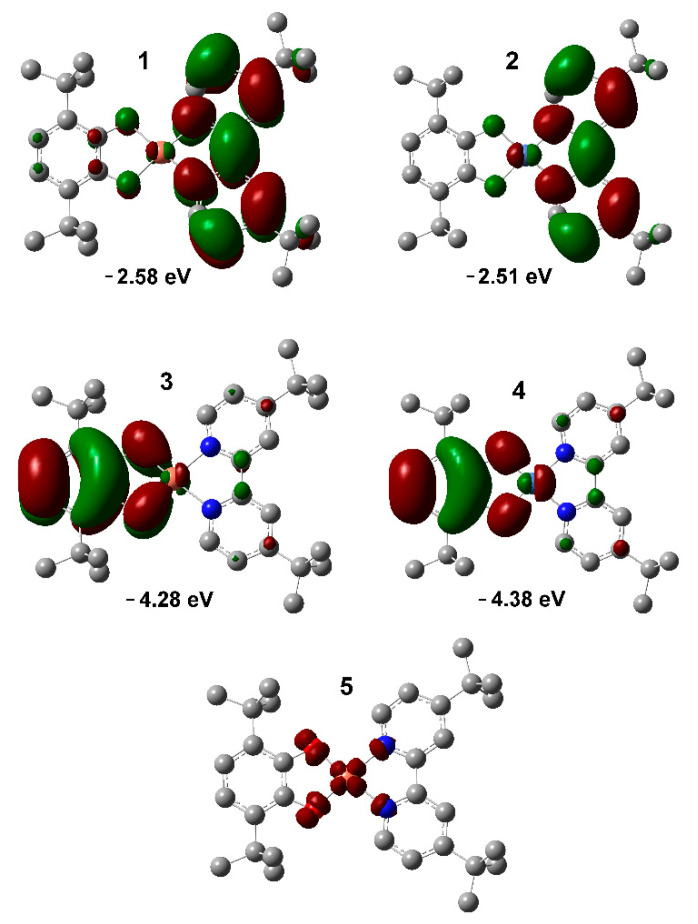
Views of calculated frontier orbitals for complexes **1** (1, 3; isovalue = 0.02), and **2** (2, 4; 1, 3; isovalue = 0.02) and the spin density distribution for complex **1** (5; isovalue = 0.005).

**Figure 10 molecules-27-08175-f010:**
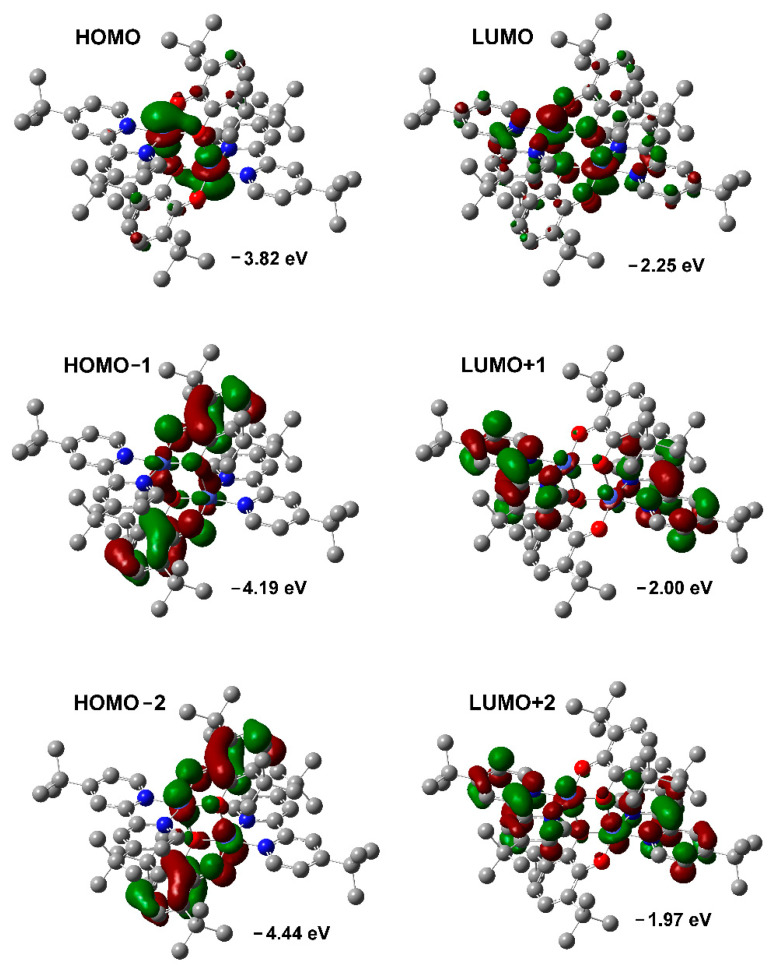
View of calculated frontier orbitals for complex **3** (isovalue = 0.03).

**Table 1 molecules-27-08175-t001:** Characteristics of a coordination polyhedron and bridging lengths for dimer **3∙toluene** and reported Cu^II^ and Co^II^ oligomers on the base of 3,5-di-*tert*-butyl-*o*-benzoquinone type ligand.

Contact	3∙Toluene	[Cu^II^(3,5-Cat)(bipy)]_2_ [47]	[Cu^II^(3,5-SQ)_2_]_2_ [97]	[Co^II^(3,5-SQ)_2_]_4_ [98]
	*dimer*	*dimer*	*dimer*	*tetramer*
M⋯M *, Å	2.8847 (7)	3.0742 (8)	3.360	3.163/3.306/3.317
M⋯O *, Å	2.0222 (18)	2.330 (2)	2.416 (4)	2.075/2.133/2.175
**Coordination polyhedron**				
τ_5_ [102] **	0.087, square pyramid	0.228, square pyramid	0.420, square pyramid	-
-	-	-	-	octahedron

Notes: *—atoms from the second part of oligomer (Symmetry transformations used to generate equivalent atoms: −x + 1, −y, −z + 1); **—calculated by the authors of this work; **[Cu^II^(3,5-Cat)(bipy)]_2_**-*bis*((3,5-di-*tert*-butyl-catecholato)(2,2′-bipyridine)copper(II)); **[Cu^II^(3,5-SQ)_2_]_2-_***bis*((*bis*(3,5-di-*tert*-butyl-semiquinonato))(2,2′-bipyridine)copper(II)); **[Co^II^(3,5-SQ)_2_]_4-_***tetrakis*((*bis*(3,5-di-*tert*-butyl-semiquinonato))(2,2′-bipyridine)cobalt(II)).

**Table 2 molecules-27-08175-t002:** Selected bond lengths for complexes **1**–**3**.

1·THF		2 (Molecule A)/(Molecule B)		3∙Toluene	
Bond	Å	Bond	Å	Bond	Å
Cu(1)-O(1)	1.860 (2)	Ni(1)-O(1)	1.8137 (14)/1.8110 (14)	Co(1)-O(1)	1.9375 (18)
Cu(1)-O(2)	1.857 (3)	Ni(1)-O(2)	1.8096 (15)/1.8118 (15)	Co(1)-O(2)/Co(1)-O(2) *	2.0552 (17)/2.0222 (18)
Cu(1)-N(1)	1.975 (3)	Ni(1)-N(1)	1.8755 (17)/1.8773 (18)	Co(1)-N(2)	2.113 (2)
Cu(1)-N(2)	1.975 (3)	Ni(1)-N(2)	1.8782 (17)/1.8768 (17)	Co(1)-N(1)	2.115 (2)
O(1)-C(1)	1.348 (5)	O(1)-C(1)	1.353 (3)/1.351 (3)	O(1)-C(1)	1.335 (3)
O(2)-C(2)	1.347 (4)	O(2)-C(2)	1.357 (2)/1.356 (2)	O(2)-C(2)	1.364 (3)
N(1)-C(13)	1.335 (5)	N(1)-C(15)	1.342 (3)/1.343 (3)	N(2)-C(24)	1.340 (3)
N(1)-C(17)	1.340 (4)	N(1)-C(19)	1.349 (3)/1.352 (3)	N(2)-C(20)	1.341 (3)
N(2)-C(18)	1.346 (4)	N(2)-C(20)	1.355 (3)/1.349 (3)	N(1)-C(19)	1.344 (3)
N(2)-C(22)	1.336 (5)	N(2)-C(24)	1.339 (3)/1.339 (3)	N(1)-C(15)	1.341 (3)
C(1)-C(2)	1.408 (6)	C(1)-C(2)	1.411 (3)/1.406 (3)	C(1)-C(2)	1.434 (4)
C(2)-C(3)	1.399 (6)	C(2)-C(3)	1.393 (3)/1.391 (3)	C(2)-C(3)	1.400 (4)
C(3)-C(4)	1.388 (6)	C(3)-C(4)	1.393 (3)/1.397 (3)	C(3)-C(4)	1.401 (4)
C(4)-C(5)	1.367 (7)	C(4)-C(5)	1.378(3)/1.383 (3)	C(4)-C(5)	1.367 (4)
C(5)-C(6)	1.397 (6)	C(5)-C(6)	1.394 (3)/1.393 (3)	C(5)-C(6)	1.392 (4)
C(1)-C(6)	1.401 (5)	C(1)-C(6)	1.392 (3)/1.396 (3)	C(1)-C(6)	1.407 (4)

Notes: *—atoms from the second part of **[Co^II^(3,6-Cat)(bipy*^t^*^Bu^)]_2_** dimer (Symmetry transformations used to generate equivalent atoms: −x + 1, −y, −z + 1).

**Table 3 molecules-27-08175-t003:** Values of “metal-heteroatom” distances in some known monomeric “α-diimine-M^II^-catecholate” type complexes.

Bond	Cu^II^(3,5-Cat)(bipy) [47]	Ni^II^(3,6-Cat)(bipy) [32]	Co^II^(3,6-Cat)(DAD) [43]
	*monomer*	*monomer*	*monomer*
M-O, Å	1.901 (5), 1.870 (5)	1.814 (3), 1.815 (3)	1.8066 (13), 1.8104 (13)
M-N, Å	1.993 (6), 1.999 (6)	1.882 (4), 1.891 (3)	1.8743 (16), 1.8711 (16)

Notes: **Cu^II^(3,5-Cat)(bipy)**-(3,5-di-*tert*-butyl-catecholato)(2,2′-bipyridine)copper(II); **Ni^II^(3,6-Cat)(bipy)**-(3,6-di-*tert*-butyl-catecholato)(2,2′-bipyridine)nickel(II); **Co^II^(3,6-Cat)(DAD)**-(3,6-di-*tert*-butyl-catecholato)(2,3-dimethyl-1,4-*bis*-(2,6-di-*iso*-propylphenyl)-1,4-diaza-1,3-butadiene)cobalt(II).

**Table 4 molecules-27-08175-t004:** The comparison of CV data for complex **2** and **Ni^II^(3,6-Cat)(bipy)**.

Compound	*E*_1/2_^Red^, V	*E*_1/2_^Ox^, V	Δ*E*_HOMO→LUMO_, eV
**2**	−2.01	−0.20	1.81
**Ni^II^(3,6-Cat)(bipy)**	−1.86	−0.21	1.65

**Table 5 molecules-27-08175-t005:** The characteristics of the CT peaks for complexes **1**–**3** in comparison with related known chromophores.

Solvent	1·THF		Ni^II^(3,5-Cat) (bipy^tBu^) [67]	2		Ni^II^(3,6-Cat) (bipy) [32]	3∙Toluene		Ni^II^(3,6-Cat^Gly^) (bipy) [32]
	λ (ε)	E	λ (ε)	λ (ε)	E	λ (ε)	λ (ε)	E	
*CH_3_CN*	501 (1.29)	2.475	-	543 (3.94)	2.284	-	657 (1.67)	1.887	-
*DMF*	523 (1.28)	2.371	554 (1.83)	560 (2.98)	2.214	580 (3.50)	674 (2.74)	1.840	618 (3.08)
*CH_2_Cl_2_*	540 (1.31)	2.296	572 (2.09)	580 (4.89)	2.138	610 (4.35)	695 (5.19)	1.784	685 (4.04)
*THF*	583 (0.87)	2.127	618 (1.95)	617 (3.98)	2.010	640 (3.43)	715 (9.36)	1.734	720 (3.18)
*Toluene*	654 (0.95)	1.896	683 (1.70)	680 (2.24)	1.824	715 (3.57)	insoluble	-	810 (3.38)
**Parameters of solvatochromic shift**									
**Δλ_(Tol–CH3CN)_**	153		-	137		-	-		-
**Δλ_(Tol–DMF)_**	131		129	120		135	-		192
**Δλ_(THF–CH3CN)_**	82		-	74		-	58		-
**Δλ_(THF–DMF)_**	60		64	57		60	41		102

Notes: **λ**—wavelength, nm; **ε**—extinction coefficient, 10^3^ M^−1^·cm^−1^; **E**—LL’CT energy, eV; **Δλ**—solvatochromic shifts within two different solvents (given in brackets), nm.

## Data Availability

The data presented in this study are available in the article and Appendix A.

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
