# Peer review of "Charge Transfer Chromophores Derived from 3d-Row Transition Metal Complexes"

_molecules, 2022, doi:10.3390/molecules27238175_

Round 1
Reviewer 1 Report
The ms. of prof. Pashanova et al. describes three similar compounds with three different metal ions. There are several methods applied.
Generally, it looks very lengthy due to extended correlation with different results related to the title three species. The synthesis seems demanding as the compounds may be water and oxygen reactive.
Before the acceptance the authors shell consider several issues stated below.
'visible' CT may be misleading, as visible is supposed to be only for a visible light absorption, not that one may see something (being visible).
The Abstract text below the mentioned structures shell correspond to obvious or principal results, not to general conclusions.
The EPR conclusions shell rely on any outside reference (not from the authors). The Co(II) EPR spectra are difficult to measure, and the absence of the signals hardly explain anything.
The authors describe the LL or L+M CT transitions in the visible light region absorption. I cannot find any relation to d-d transitions typical to this area of spectra for the late transition metal species. Are d-d bands not present there? There is mentioned 400-500 nm region of important investigation, but the peaks in the figure are at 400-800 area ?
UV-Vis-NIR shell be written everywhere, as Vis a shortened word, while UV and NIR are abbreviations.
The spaces in the references shell be checked thoroughly; especially between the name of a metal and its oxidation state, where it shell be omitted. Also, some journal capitals.
Reviewer 2 Report
Manuscript Piskunov et al. describes new CT chromophores based on 3d-transition metal complexes with identical ligand environments derived from 4,4'-di-tert-butyl-2,2'-bipyridyl and 3,6-di-tert-butyl-o-benzoquinone. The close structure of the complexes made it possible to reveal the relationship between the photophysical behavior and the nature of the metal atom. I consider that the manuscript can be published in Molecules after response on following minor remarks: 1) Although the introduction is too detailed, the authors do not comment on the choice of di-tert-butyl-substituted 2,2ʼ-bipyridyl as a acceptor moiety. Evidently, the introduction of donor tert-butyl in acceptor will increase the HOMO-LUMO gap and shift the absorption band hypsochromically. What is reason of this choice? 2) Figure 1 begins with the synthesis of copper complex 1, however the discussion begins with a description of the synthesis of nickel complex 2. It's inconsistent. 3) The description of known works should be corrected. For example, “Chang with co-workers have shown [67],” will be better than “The authors have shown [67],” in line 426. 4) Some typos should be corrected. For example: a) “the doctor and the acceptor” in line 559 b) “from toluene to DMF for 1, 2; from THF to DMF” in line 664. Shold be “from toluene to MeCN for 1, 2; from THF to MeCN”
